# Autoimmune Thyroid Disease in Patients with Down Syndrome—Review

**DOI:** 10.3390/ijms26010029

**Published:** 2024-12-24

**Authors:** Weronika Szybiak-Skora, Wojciech Cyna, Katarzyna Lacka

**Affiliations:** 1Student’s Scientific Society, Poznan University of Medical Sciences, 60-355 Poznan, Poland; weronikaszybiak@gmail.com (W.S.-S.); cynawojtek2@gmail.com (W.C.); 2Department of Endocrinology, Metabolism and Internal Medicine, Poznan University of Medical Sciences, 60-355 Poznan, Poland

**Keywords:** trisomy 21, autoimmunity, thyroid disease, Hashimoto’s thyroiditis, Graves’ disease

## Abstract

Down syndrome develops due to the presence of supernumerary chromosome 21. This diagnosis is made in approximately 1:800 live births. The tendency to develop autoimmune disorders like idiopathic arthritis, celiac disease, diabetes mellitus type 1, vitiligo and autoimmune thyroid disease is strongly expressed in patients with Down syndrome. Autoimmune thyroid diseases consisting of Hashimoto’s thyroiditis and Graves’ disease are specifically prevalent in patients with Down syndrome. The aim of our study is to collect available data connecting the pathogenesis and clinical course of autoimmune thyroid diseases in patients with Down syndrome of different ages and compare them to control groups. According to published data, the incidence ratio of Hashimoto’s thyroiditis diagnosis in patients with Down syndrome is elevated compared to in age-matched controls without this chromosomal aberration, similarly to Graves’ disease risk, which is also increased in a group of patients with Down syndrome. What is more, both Hashimoto’s thyroiditis and Graves’ disease are diagnosed at an earlier age than in the healthy population and are not correlated with gender or a family history of autoimmune diseases.

## 1. Introduction

Down syndrome (DS) is one of the most well-known disease entities, the symptoms of which can be evident from the first moments of life. The incidence of this chromosomal aberration is estimated to be 1:800 live births [1]. Nowadays, a preliminary diagnosis can be made prenatally (the obtaining of the karyotype via amniocentesis or chorionic villus sampling, cell-free DNA or ultrasound findings), making it possible to undertake appropriate therapeutic management from the first days of a newborn’s life. Appropriate diagnosis and treatment, undertaken at an early stage, allow for the better control of disorders that co-occur with DS [2]. Nowadays, a broader understanding of trisomy of the 21st pair of chromosomes (T21) and its complications has led to an increase in life expectancy to an average of 60 years of age for individuals with DS [3].

Not only is DS characterized by phenotype presentation, like low-seat ears, a flat nasal bridge or epicanthic folds, but it also has an impact on different health disturbances which strongly affect nervous, cardiovascular, respiratory, digestive and even musculoskeletal systems [4]. One of the most characteristic features is that individuals with DS present with a strong susceptibility toward autoimmune disorders like idiopathic arthritis, celiac disease, diabetes mellitus type 1 (DM1), vitiligo, alopecia and thyroid disfunction: Hashimoto’s thyroiditis (HT) and Graves’ disease (GD) [5,6,7].

HT, called autoimmune thyroiditis, is a disease with autoimmune pathogenesis, which most commonly leads to a decreased serum concentration of thyroid hormones. The pathogenesis of HT remains uncertain; however, to current knowledge, genetic, epigenetic and environmental factors are known to play a key role in triggering the development of HT [8]. Endogenous factors, such as intestinal microflora, may also influence the development of autoimmune thyroid diseases. It has been proven that an imbalance in the intestinal microflora can lead to a lower concentration of hydrogen sulfide, which alleviates the severity of thyroiditis [9].

In the course of HT, the infiltration of lymphocytes, including lymphocyte types B and T, in the thyroid parenchyma and the thyroid gland’s volume are mostly increased [10]. In the pathogenesis of HT, both humoral and cellular immunity are involved [11,12]. Humoral disorders present as an increased serum concentration of specific antibodies: anti-thyroid peroxidase (anti-TPO) and anti-thyroglobulin (anti-TG) [11,13]. T-cells are responsible for homeostasis disturbance and for starting the autoimmune process against the thyroid gland [14]. CD8+ T-cells against thyroid antigens were found in patients suffering from HT. It is assumed that in HT, there is a dysfunction of T-suppressors and T-regulatory cells [11,15]. It has also been suggested that small tissue extracellular vesicles (sEVs), which contain miRNAs and can transfer them to other tissue, may be involved in HT pathogenesis pathways and may provide a link between T-cell dysregulation and inflammation in thyroid cells. A study by Li et al. found that the transfer of specifically miR-142-3p from T-regulatory cells to thyrocytes via sEVs resulted in the development and progression of HT. Increased expression of miR-142-3p not only leads to T-regulatory cell dysfunction but is also responsible for the maintenance of inflammation in thyroid cells in patients with HT [16]. Moreover, the concentration of the IL-17, IL-22 or IL-12 cytokines has been shown to be increased in patients with HT [17,18]. The prevalence of HT diagnosis in patients with DS is elevated at 13–34% [5,19,20] compared to in age-matched patients without this chromosomal aberration, for whom the incidence is 1.3% [21].

The prevalence of GD diagnosis is also increased in patients with T21 [5]. Epidemiological data on the susceptibility of children with DS to GD indicate that patients with this chromosomal aberration present a 6.5‰ risk of development [22] compared to healthy controls, at 1.07‰ [23]. GD is an autoimmune disorder which is commonly associated with hyperthyroidism [5]. The pathophysiology of GD is connected with the presence of circulating anti-thyroid-stimulating-hormone receptor antibodies (TSH-R), triggering thyroid cells and leading to thyrotoxicosis. Similarly to HT, both humoral and cellular immunity are involved in GD development [24].

The prevalence of autoimmune thyroid diseases (AITDs) in patients with T21 is not correlated with gender, which distinguishes them from AITDs occurring among the healthy population, in which the presence of thyroid conditions is correlated with female gender [5,25,26].

The aim of our study was to collect available data connecting the pathogenesis and clinical course of AITD in patients with DS of different ages and compare them with the population without chromosomal aberrations.

## 2. Study Selection Process

After title/abstract screening, the authors included original articles as described below. Other review papers, meta-analyses and case reports mentioned in this article were used solely to present a summary of the discussed matter. We included papers with information about the pathogenesis of autoimmunity in the thyroid gland and the clinical presentation of HT and GD, especially focused on patients with T21. All articles not meeting the inclusion criteria were excluded by both authors. We decided to exclude articles in languages other than English/Polish, articles for which only the abstract was available, articles published prior to 2000 and articles without information about the pathogenesis of HT and GD. The study selection process is shown in Figure 1.

## 3. Down Syndrome—Pathogenesis and Clinical Implementation

DS develops as a consequence of the presence of additional chromosome 21. In patients with DS, we can distinguish free trisomy 21 in 95% of patients and translocation trisomy 21 in 5% of cases, where translocation is usually connected with t(14;21) or t(21;21) and mosaicism in 2% of cases [4,27,28]. Free T21 develops as a consequence of an error during maternal (87%) or paternal meiosis, or uncommonly in the course of mitosis after the formation of the zygote [4].

Risk factors affecting DS include advanced maternal age, which appears to be the main risk factor. Advanced maternal age affects the process of oocyte formation and leads to impaired meiosis [29]. In addition, studies have shown that the mother’s socioeconomic status may be related to the second stage of meiosis, which is associated with exposure to tobacco, emotional stress or folic acid deficiency [30,31].

Some clinical features of DS can be observed at a prenatal stage. In ultrasonography, subsequent characteristic features can be found: increased nuchal translucency, the absence of a nasal bone, an increased frontomaxillary angle, tricuspid valve regurgitation and absent or reduced flow in the ductus venosus [32]. The detection of abnormalities during ultrasonography constitutes an indication that more sensitive and specific tests are needed, like the obtaining of the karyotype via amniocentesis or chorionic villus sampling and cell-free DNA [3,33]. Pregnancy affected by DS always needs medical and psychological care due to a higher risk of miscarriage after 12 weeks—at a rate of approximately 30%. The risk of spontaneous fetal death is positively correlated with the mother’s age [34].

DS diagnosis is strongly correlated with numerous health problems and the need for multifocal medical care. T21 increases the prevalence of sleep apnea, early-onset Alzheimer’s disease, hearing loss or cataracts [2]. Most of the diseases associated with DS are shown in Figure 2 below.

Moreover, the tendency to develop immune system disorders is strongly expressed in patients with DS. Chromosomal aberration predisposes them to infections, which occur more frequently than in the healthy population and have a more severe course [47]. Also, the prevalence of autoimmune diseases is increased in this group of patients. They are more prone to celiac disease, which can lead to multiple deficiencies and, as a result, developmental disorders, especially in the pediatric population [48]. Other autoimmune disorders that are more common in the population with DS include DM1 or alopecia, but autoimmune disorders related to the thyroid gland are most commonly seen in this group of patients. AITDs include HT and GD. Interestingly AITDs have no correlation with the sex or family history of patients with DS, which differs from the healthy population [49]. What is more, the onset of AITDs is earlier in the group of patients with DS than in controls [25]. The frequency of thyroid disfunction in patients with DS indicates that the control of symptoms and the assessment of the serum concentration of TSH and thyroid hormones are recommended [50].

## 4. Pathogenesis of Autoimmune Thyroid Disorders in Patients with Down Syndrome

DS, defined as T21, is a well-known genetic condition that occurs with a higher incidence ratio of numerous autoimmune diseases. Adrenal insufficiency, thyroid disorders and DM1 are only a few of the possible manifestations of autoimmunity in patients with DS [51]. The mechanism of autoimmunity in T21 patients remains uncertain. Nevertheless, there are known predisposing factors occurring with DS that impact the pathogenesis of autoimmunity in these individuals.

The AIRE (autoimmune regulator) gene is located on chromosome 21, at position q22.3 [52,53]. In animal studies, AIRE has been observed to affect the expression of tissue-restricted antigens (TRAs) in medullar thymic epithelial cells, controlling antigen presentation, maturation and the production of chemokines [53]. TRA expression is fundamental for proper thymocyte development, playing a key role in the negative selection of thymocytes; when disrupted, it may lead to autoimmunity. Autoimmune regulatory gene expression on both mRNA and protein levels differs in studies [52,53]. Nevertheless, the effect on immune dysregulation in patients with DS is clear. Interestingly, Giménez-Barcons et al. demonstrated a correlation between a lower concentration of AIRE and patients that developed hypothyroidism, supposedly of autoimmune origin. In addition, one patient was diagnosed with GD. Thus, the correlation between AIRE and AITD is worth further research [48].

Patients with DS present histopathologic thymus abnormalities. Most importantly, thymic tissue in patients with DS reveals hypoplasia and altered architecture cortical atrophy with a loss of corticomedullary demarcation [52,53].

T21 is characterized by elevated pro-autoimmunity inflammatory cytokines, such as interleukin 17 (IL-17), interleukin 22 (IL-22) and TNF-α (tumor necrosis factor alpha) [53]. Adults with DS have a high CD8+/CD4+ lymphocyte ratio, which is defined by a higher population of CD8+ T-cells which excessively produce TNF-α [35,53]. Studies have revealed the hypersensitivity of T-cells to IFN-α stimulation and defined it as a positive factor toward autoimmunity in DS [54]. What is more, conventional CD 4+ T-cells overproduce IL-17 and IL-22, which is especially important according to the impact of IL-17 on development in autoimmune diseases with a high concentration of type I IFN signaling [53]. The titer of regulatory CD 4+ T cells (Treg) is elevated; nonetheless, their function is flawed. Tregs leave thymopoiesis with defects alongside decreased T-cell receptor excision circles (Trec) found in DS, contributing to an increase in the incidence of autoimmune diseases in patients with DS [35]. What is more, the chronic activation of IL-6 signaling and a higher ratio of plasmablasts and CD11c^+^Tbet^high^CD21^low^B cells are found in individuals with DS—promoting autoimmunity [55].

Kaczmarek et al. revealed that the apoptosis process plays a crucial role in HT pathogenesis. Apoptosis marker concentration, like caspase-3, was statistically significantly higher in the group with HT than in healthy controls. What is more, the activity of CD20 and CD43 lymphocytes was increased in the group with HR [56]. According specifically to HT, the cytotoxic T-lymphocyte-associated protein 4 gene (CTLA-4) +49A/G has an autoimmune-promoting effect. CTLA-4 is a molecule found on T cells. The production of malfunctioning receptors leads to a reduction in CTLA-4’s inhibitory function on T-cell activation, which promotes the development of HT [57]. Interestingly, research revealed that the AA genotype is considered a protective factor in the pathogenesis of HT in children with DS [57].

All the above-mentioned factors promote autoimmunity in the population with DS, as summarized in Figure 3.

What is more, not only patients with DS but also patients with other genetic disorders are predisposed to thyroid abnormalities, such as those with Turner syndrome (TS) or Klinefelter syndrome [58,59]. According to Sagad Omer Obeid et al., thyroid autoimmunity occurs approximately in 38% of TS cases, of which 12.7% present clinical hypothyroidism and 2.6% have clinical hyperthyroidism [60].

## 5. Thyroid Function Screening in Patients with Down Syndrome

In compliance with the American Academy of Pediatrics, the thyroid function of individuals with DS should be closely monitored. TSH concentration should be checked at birth and at 6 and 12 months of age, followed by annual TSH measurement afterwards [61]. In the case of abnormal TSH, fT4 concentration should be performed [62]. The detection of autoimmunity is based on the presence of anti-thyroid antibodies. What is more, if antibodies were detected earlier, TSH measurement should be performed every 6 months, and “watchful observation” must be applied [61]. Moreover, the levels of vitamin D should be closely monitored. As one of the environmental factors in autoimmune thyroiditis, vitamin D deficiency leads to a proinflammatory state [8].

## 6. Hashimoto’s Thyroiditis—Characteristics in Down Syndrome

The most common cause of hypothyroidism is HT [63]. The clinical manifestation of HT can involve the presentation of local symptoms associated with an increased volume of the thyroid gland. Patients may present symptoms such as difficulty swallowing due to esophageal compression, shortness of breath due to airway compression, or voice tone abnormalities due to recurrent laryngeal nerve compression. Systemic symptoms most commonly include the typical clinical features of hypothyroidism; nevertheless, patients may present symptoms of hyperthyroidism or remain euthyroid [63,64,65]. The clinical manifestation of hypothyroidism in individuals with DS can be challenging. Many symptoms are very similar to T21 characteristic features, such as hypotonia, dry skin or slower mental development [66,67]. Thus, the constant monitoring and early management of HT with levothyroxine is crucial [67]. Subclinical hypothyroidism treatment in individuals with DS is debatable, and close monitoring should be conducted [68]. Patients with HT present not only clinical symptoms but also changes in laboratory test results. An increased serum concentration of specific antibodies—anti-thyroid peroxidase (anti-TPO) or anti-thyroglobulin (anti-TG)—and disbalance in the serum concentration of TSH and thyroid hormones can be present [11,13]. Early treatment is essential in view of the finding that untreated hypothyroidism in patients with T21 that may lead to slower mental development, hypotonia, anemia or altered cardiac function [68,69]. In terms of HT in DS, a multidisciplinary approach must be present, including not only physicians but families of patients with DS as well. Appropriate encouragement of families to closely monitor and observe individuals with DS is important to preserve proper treatment and constant monitoring [61].

HT is the most common autoimmune disease accompanying T21 [6]. The development risk of HT is increased among patients with trisomy compared to the non-trisomy population [19,20].

Abdulrazzaq et al. investigated the occurrence of autoimmune diseases in patients with DS. A total of 25 (27.2%) out of 92 patients presented one autoimmune disease, and 1.1% suffered from two coexistent autoimmune disabilities. The risk of DM1 and hypothyroidism development was strongly correlated with DS; in contrast, celiac disease presented the least association with DS [70]. Another study, presented by Pepe et al., proved the influence of the autoimmune process in thyroid function deterioration and the evolution of subclinical hypothyroidism into overt hypothyroidism in a group of patients with T21 [71].

Moreover, one investigation, designed by Giménez-Barcons et al., demonstrates a high incidence ratio of autoimmune thyroid disorders in a group of patients with DS; 52.6% of individuals with DS developed hypothyroidism connected with autoimmunity. In the non-DS group of controls, no autoimmune thyroid dysfunction was observed. Authors have also investigated the role of AIRE, the transcription factor of which is located on chromosome 21. In patients with DS, a lower concentration of AIRE has been observed, which indicates the genetic etiology of autoimmune disbalance [48].

The results of investigations that aimed to present the increased prevalence of HT in patients with DS are presented in Table 1 and are graphically presented in Figure 4. Based on the mentioned studies, patients with DS are more susceptible to HT in comparison to healthy individuals.

Patients with DS also present a different clinical and biochemical course of HT. The onset of HT begins earlier in the population with trisomy than in controls (mean age of 6.5 years old) [5,19]. Biochemical presentation, in contrast to the group of non-DS patients in which HT most commonly presents as euthyroidism, in patients with chromosomopathies (DS or TS) presents subclinical hypothyroidism in most cases, followed by hypothyroidism [71,77].

Variances in the biochemical pattern of HT in patients with T21 may be the result of a non-immune thyroid disorder that manifests as increased TSH serum concentration without changes in hormone serum concentration and clinical symptoms, present in the first years of life. In 70% of cases, this is a transient disorder that resolves spontaneously, but the presence of antibodies reduces the chance of spontaneous remission [78]. A couple of studies have evaluated the several-year follow-up of patients with DS who developed HT. Most patients showed a progressive deterioration of thyroid function. Individuals with DS were more likely to develop subclinical hypothyroidism, overt hypothyroidism or hyperthyroidism compared to the non-trisomy population, where euthyroidism was more often found [25,79].

Zwaveling-Soonawala et al. investigated the influence of supplementation of thyroxin during the first years of life in the course of thyroid dysfunction in a group of patients with DS. The authors proved the decreased development of autoimmunity against thyroid tissues in patients with thyroxine supplementation compared to patients with DS without thyroxine supplementation; however, these results were not statistically significant. Nevertheless, the obtained results may indicate the protective effect of applying early thyroxine treatment [72].

In addition, patients with trisomy and HT are more susceptible to GD conversion. Aversa et al. demonstrated that in patients with chromosomal nondisjunction (both DS and TS), the incidence of GD preceded by HT was 25.7%, proving the link between chromosomal aberration and an increased risk of HT progression to GD [79]. The differences between the course of HT in patients with DS and the non-DS group is presented in Table 2.

### 6.1. Hashimoto’s Encephalopathy in Patients with DS 

Another matter worth mentioning in HT is Hashimoto’s encephalopathy (HE). HE is a poorly understood disease for which diagnosis remains difficult. In adults, HE manifests with consciousness disturbance, memory loss, sleep disturbance and seizures, as well as psychiatric symptoms including behavioral changes [80]. The high incidence of HT in the group of patients with DS suggests careful monitoring for HE. Described mostly in case reports, patients with DS in the course of HE presented functional decline, behavioral changes and myoclonic jerks accompanied with abnormal EEG [81,82]. In children with DS, autistic regression can be observed—manifesting as a loss of speech, becoming dependent in daily-life activities and involuntary sudden movements [81,83]. No specific diagnostic parameter for HE has been found; however, the anti-TPO titer remains a valuable factor [80]. Despite the wide spectrum of clinical evaluations of HE, corticosteroid treatment gives satisfactory results in most cases [82,84]. Due to the lack of epidemiological studies concerning HE in the population with DS, this diagnosis should always be considered given the high incidence ratio of autoimmune thyroiditis in patients with Down syndrome.

### 6.2. Cardiac Findings in Patients with DS with Hypothyroidism

Apart from thyroid abnormalities, the malfunction of the cardiovascular system is a major health problem among the population with DS, especially congenital heart defects (CGHs), such as atrioventricular septal defects (AVSDs), ventricular septal defects (VSDs) or patent ductus arteriosus (PDA) [85]. The prevalence of CGH is estimated at around 40% in the population with DS [86,87].

An interesting study was conducted by Suzen Celbek et al. in order to investigate the impact of hypothyroidism on cardiac findings in patients with DS below 15 years of age [87]. The authors assumed that there is a correlation in which hypothyroidism negatively affects cardiac systolic functions in the population of children with DS. What is more, intraventricular septum and left ventricle posterior wall thicknesses are increased in children with DS regardless of hypothyroidism. This finding suggests that hypothyroidism (in this case, subclinical) has no effect on cardiac morphology, and is rather an effect of DS on the myocardial structure [88].

## 7. Graves’ Disease—Characteristics in Down Syndrome

AITD is not only associated with hypothyroidism and HT but can also present as hyperthyroidism in the form of GD. This autoimmune disorder is characterized by the manifestation of an increased concentration of TSH-R antibodies, which have an excitatory effect on the receptor for TSH [5]. The interaction of the antibodies with the receptor results in the development of thyrotoxicosis and the enlargement of the thyroid gland [24].

The common manifestations of GD include hyperthyroidism symptoms, such as tachycardia, atrial fibrillation, hypertension, warmed and moist skin, menstrual disorders, heat intolerance and fatigue, but also indicators of increased metabolic processes such as concurrent increased appetite and weight loss [89]. The triad of symptoms characteristic of GD, called the Merseburger triad, consists of thyrotoxicosis, diffuse goiter and ophthalmopathy (orbitopathy) [90]. Ophthalmopathy is one of the most common extrathyroidal GD manifestations [91]. The pathogenesis of orbitopathy is related to the presence of TSHR expression in orbital tissues. Increased TSHR activation, associated with the presence of antiproteins, increases the differentiation of orbital preadipocytes into adipocytes, resulting in the proliferation and an increased volume of orbital tissues [92,93]. The Clinical Activity Scale (CAS) is used to assess orbitopathy in GD. Based on the CAS, the activity of orbitopathy is assessed, for which the cutoff point is defined as ≥3/7 [94].

GD in patients with DS is mostly symptomatic; however, similarly to HT in DS, GD may be difficult to determine on the basis of clinical symptoms [95]. Due to the overlapping symptoms of T21 itself and GD, a close diagnostic approach must be routinely used. Typical symptoms for both GD and DS include increased irritability and behavioral disturbance [4,95].

The prevalence of GD diagnosis is also increased in patients with T21 [5]. Epidemiological data on the susceptibility of children with DS to GD indicate that patients with T21 present a 6.5‰ risk of development [22] compared to healthy controls, at 1.07‰ [23]. Some studies show an upward trend in the incidence of GD with the age of patients with T21. This can be compared to the study by Tüysüz et al., in which children aged 0–10 were evaluated, and no case of hyperthyroidism was identified [96], while four patients were identified in the study by Calcaterra et al., in which older children and adolescents were evaluated [96]. The course of GD in patients with DS is not different than in controls, but it is more often associated with other autoimmune diseases. Moreover, De Luca et al. presented the results of their study in which 6 out of 28 patients with DS demonstrated a conversion from HT to GD, indicating that DS may increase the risk of developing HT-based GD in patients with T21 [49].

According to a longitudinal study by Dos Santos et al., every individual with DS with hyperthyroidism acquired autoimmune etiology. Interestingly, no correlation between GD and gender was found in the group of patients with T21 in comparison to non-DS patients in which the prevalence in women is clear [97].

As for the clinical presentation of ophthalmopathy in a group of patients with DS suffering from GD, the clinical data remain unclear. Goday-Arno et al. demonstrated that 3 of 12 patients presented ophthalmopathy. What is more, authors indicated that ophthalmopathy occurs at a younger age in patients with DS compared to controls, but the frequency of this clinical symptom was decreased in the group with trisomy [22]. In contrast to the previous article, a study established by Nurcan Cebeci et al. presented 161 children with GD, among whom 13 were suffering from DS. Ophthalmopathy was not diagnosed in any patients with DS [95].

Four studies mentioning GD in groups with DS are summarized and presented in Table 3. According to previous results, GD prevalence in the population with DS is rare but more common in comparison to the healthy population mentioned in the studies [20]. What is more, no correlation between gender and GD in DS has been found [95]. Interestingly, T21 may be a promoting factor for conversion from HT to GD [49].

No consensus regarding the treatment of GD in patients with DS is present. Nevertheless, based on the mentioned studies, treatment with anti-thyroid drugs (ATDs) (methimazole) remains the most popular method of management of GD in the population with T21 [49,95,97]. In contrast, Goday-Arno et al. presented a study in which no patient achieved remission after the usage of ATDs and definitive therapy with I^131^ in the management of GD [22]. Thyroidectomy in patients with DS is avoided and should be performed only in the case of a thyrotoxic state that must be terminated rapidly or in the case of unsuccessful treatment of other kinds [22,99]. Early treatment is essential to successfully prevent patients from developing typical complications of untreated GD such as tachyarrhythmias (most commonly, atrial fibrillation) or severe orbitopathy that may require glucocorticosteroid treatment or surgical involvement [100,101]. According to Hoang et al., all patients starting on ATD therapy should be closely monitored every 2–4 weeks by obtaining serum TSH, fT4 and fT3. After a euthyroid state is achieved, the dosage of ATD should be lowered, and monitoring is advised to be performed in 4–6 weeks. A euthyroid state should be achieved with the lowest (effective) dosage of ATD. Afterwards, laboratory testing is advised in 2–3 months [102]. As in many other diseases occurring in DS, special care must be given to the patient and to the patient’s family [61]. What is more, a multidisciplinary group of physicians is required considering the complexity of GD symptoms [102].

### Graves’ Disease in Patients with Down Syndrome and Occurrence of Moyamoya Disease

Moyamoya disease is a rare vasculopathy, in the course of which the formation of net-like vessels occurs. Impaired vessels develop in the lateral basal regions of the cerebral areas and lead to ischemic episodes [103]. The characteristic presentation of Moyamoya disease is the bilateral hypoplasia of the internal carotid arteries, which is described in angiographic images as “a puff of cigarette smoke”. This rare syndrome may be associated with autoimmune disease, aplastic anemia, Fanconi anemia and tuberculosis, but also with Down Syndrome and thyrotoxicosis [104].

Moyamoya’s coexistence with GD has been reported in 84 cases, and 3.8% of patients also presented DS [105]. An increased concentration of thyroid hormones affects the endothelium of vessels and smooth muscles and promotes the atherosclerotic process. Moreover, thyroxin enhances the activity of the sympathetic nervous system, which correlates with the progression of Moyamoya disease [105]. In patients with DS, the concentration of endostatin is increased, which results in the suppression of angiogenesis and promotes vascular branching and the development of abnormal vascular formation [106]. As previous studies have proven, the mean age of Moyamoya disease presentation was lower (15.6 y.o) in the group of patients with both DS and GD than in patients with only GD (31.4 y.o) [105].

Moyamoya disease is an extremely rare condition, but it should be taken into consideration especially in patients with coexisting DS and GD. In this group of patients, the control of thyroid function and the maintenance of a proper concentration of thyroid hormones can decrease the risk of ischemic episodes and slow the progression of vascular disease.

## 8. Discussion

Patients with T21 are a group at particular risk of developing metabolic disorders [107]. Initially, due to symptomatic hypotonia and swallowing problems, infants with T21 are at risk of malnutrition and nutrient deficiencies [108]. A special group of infants with DS are those with heart defects, especially atrioventricular septal defects, who are more likely to need enteral feeding or a gastrostomy to ensure a proper calorie supply [109]. At later developmental stages, children with DS present difficulties with food bite formation and proper swallowing, and adaptation to new types of food is delayed in comparison to the healthy population [110]. Often, high-calorie products, in the form of paps, replace a balanced diet rich in fresh fruit and vegetables [110]. Furthermore, it has been proven that children with DS show a reduced sense of satiety after a meal and may eat until the plate becomes empty [111].

Children and adults with DS with obesity present problems not only with excessive calorie intake but also with a variety of disease entities that can affect metabolism [110]. Hypothyroidism is an important problem in this group of patients, especially given the greater predisposition to Hashimoto’s disease, which presents more often as subclinical or clinically overt hypothyroidism in these individuals in comparison to the healthy population [25].

Thyroid hormones influence the regulation of metabolic processes by interacting with receptors in both the white and brown fat tissues. Proper thyroid hormone concentrations also regulate normal fat distribution with a reduction in abdominal obesity, which is associated with increased cardiovascular risk [112]. Furthermore, studies indicate that leptin secreted in excess by adipose tissue cells in individuals with obesity affects the hypothalamus–pituitary axis and directly affects cells of the anterior pituitary lobe, thus disrupting the diurnal rhythm of TSH secretion [113,114]. Therefore, in patients with DS, it is extremely important to maintain a normal concentration of thyroid hormones, which have a direct impact on weight reduction. However, maintaining a healthy body weight and reducing leptin production by adipose tissue can have a positive effect on thyroid function and clinical and biochemical improvements in thyroid hormones [115].

On the other hand, patients with DS are a group particularly vulnerable to the presence of celiac disease. According to a 2018 meta-analysis, the risk of celiac disease in patients with DS is 5–8%, in comparison to the risk in the healthy population, which is 0.5–1% [116]. Celiac disease may manifest itself as a number of different symptoms in both the digestive tract and other systems. Damage to the intestinal epithelium and associated malabsorption may lead to nutritional deficiencies, weight loss and various clinical implications [116]. Creo et al. presented a patient with DS with both celiac disease and GD. The patient was diagnosed with GD based on typical endocrine abnormalities, the presence of antibodies and ophthalmopathy. The patient was treated with metamizole, and then, due to the clinical and biochemical determinants of thyroid insufficiency, l-thyroxine was administered. In the following years, the patient was diagnosed with celiac disease, the first symptom of which was menstrual disorders. The use of a gluten elimination diet resulted in the regulation of menstrual cycles and allowed for the better control of thyroid function, which led to a reduction in the dose of l-thyroxine [117]. Patients with DS have an increased risk of autoimmunity, so it is important to be aware of the possibility of several co-morbidities when controlling metabolic parameters. In 2008, the case of a boy diagnosed with DM1, celiac disease and HT was described. Treating the individual disease entities and controlling their interactions allowed the child to maintain a healthy metabolic profile [118].

Metabolic control and the maintenance of a healthy body weight in patients with DS is crucial, due to the increased cardiovascular risk and the increased risk of autoimmune diseases such as DM1 and celiac disease in this group of patients. The prevention of overweight or obesity and ensuring an adequate supply of minerals are particularly important in the context of autoimmune thyroid diseases, which are the most common autoimmune diseases in patients with DS.

## 9. Conclusions

In conclusion, thyroid autoimmunity is highly prevalent in the population with DS. In comparison to healthy controls, individuals with T21 more commonly present AITD.

HT is the most common manifestation of AITD in DS. Biochemically, HT in T21 begins earlier than in healthy controls and most commonly starts with subclinical hypothyroidism that may evolve to overt hypothyroidism because of the presence of anti-thyroid antibodies (anti-TPO, anti-Tg). If needed, early thyroxine treatment in DS is recommended, thus preventing autoimmunity development. HE is a rare manifestation of HT; however, it should not be omitted from consideration in the population with DS.

GD is a rare complication of thyroid autoimmunity. In the population with DS, the prevalence of GD is higher than in healthy controls; nevertheless, this is an uncommon finding. GD usually manifests as hyperthyroidism. Interestingly, typical ophthalmopathy is unlikely to appear in T21. What is more, no correlation between GD in DS and gender has been found. Individuals with DS with HT should be closely monitored for the transition to GD.

Moreover, the control and maintenance of a proper serum concentration of thyroid hormones are important in the name of the metabolic profile of patients with DS, who are more predisposed to overweight and obesity. The control of thyroid function affects the metabolism process and prevents an increase in weight, but on the other hand, maintaining a healthy weight in patients with DS influences the better control of thyroid function.

## Figures and Tables

**Figure 1 ijms-26-00029-f001:**
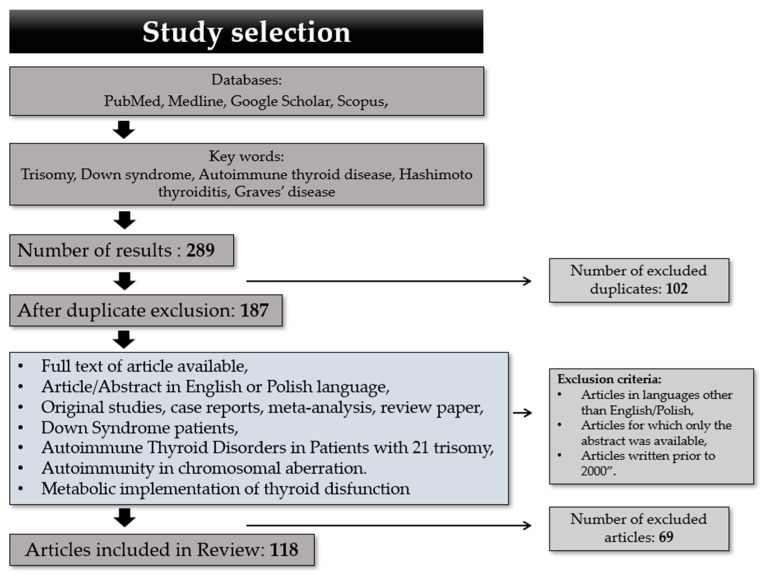
Study inclusion process.

**Figure 2 ijms-26-00029-f002:**
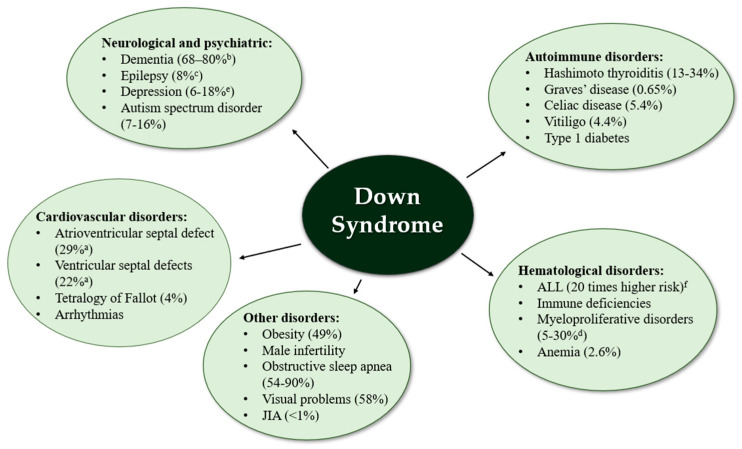
Diseases associated with trisomy 21. JIA—juvenile idiopathic arthritis; a—of all congenital heart defects; b—patients with DS aged > 65 years old; c—of children with DS; d—transient myeloproliferative disorder in patients with DS up to 3 months of age; e—in adults with DS; f—in comparison to non-DS [35,36,37,38,39,40,41,42,43,44,45,46].

**Figure 3 ijms-26-00029-f003:**
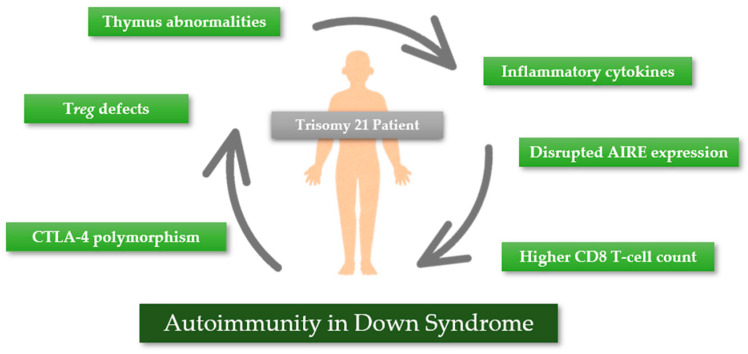
Processes leading to autoimmunity in DS. AIRE—autoimmune regulator gene; CTLA-4—cytotoxic T-lymphocyte-associated protein 4 gene.

**Figure 4 ijms-26-00029-f004:**
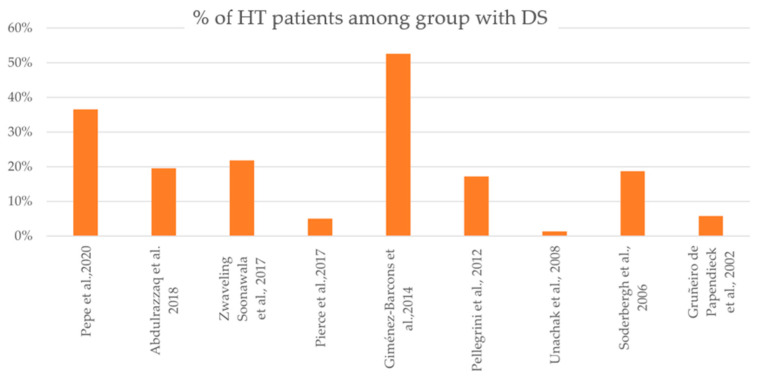
Percentage of patients with HT among the group with DS in each study, based on Table 1. DS—Down syndrome; HT—Hashimoto’s thyroiditis [35,48,70,71,72,73,74,75,76].

**Table 1 ijms-26-00029-t001:** HT occurrence in patients with DS.

Author, Year	N of DS	N Control Group	Age of DS (Range or Mean)	HT in DS	HT in Control Group	% of HT in DS	TSH	fT4	Anti-TPOAb	Anti-TGAb	*p*
Pepe et al., 2020 [71]	101	64-non-autoimmune 37-autoimmune SH	2–17 ^h^	37	-	36.6%	8.5 ± 5.5 UI/mL	14.9 ± 8.0	−	−	-
Abdulrazzaq et al., 2018 [70]	92	-	-	18 ^a^	-	19.6% ^a^	-	-	+	+	-
Zwaveling Soonawala et al., 2017 [72]	181 ^b^	-	10.7 (mean)	26 ^f^	-	21.9% ^f^	6.8 U/L ^b^	18.6 pmol/L ^b^	+	−	-
Pierce et al., 2017 [73]	565	-	0.05–28	25	-	5.1%	11.6 mIU/ml ^c^	1.03 ng/dl ^c^	+	+	<0.005 ^d^
Giménez-Barcons et al., 2014 [48]	19	21	0–10	10	0	52.6%	-	-	−	+	-
Pellegrini et al., 2012 [35]	29	29	1.4–22.8	5	0	17.2%	-	-	+	−	-
Unachak et al., 2008 [74]	140	-	0–14	2	-	1.4%	-	-	+	−	-
Soderbergh et al., 2006 [75]	48	-	11–56	9 ^e^	-	18.7% ^e^	-	-	+	−	-
Gruñeiro de Papendieck et al., 2002 [76]	137	66	0.04–16	8 ^e^	0 ^i^	5.8% ^e^	141.1 ± 98 mU/L ^e^ 4.7± 2.8 mU/1 ^g^	21.9 ±12.9 nmol/L ^e^	+	−	<0.01 ^g^

DS—Down syndrome; HT—Hashimoto’s thyroiditis; SH—subclinical hypothyroidism; ^a^—hypothyroidism; ^b^—at trial entry; ^c^—subclinical hypothyroidism; ^d^—difference in TSH and fT4 between subclinical and overt hypothyroidism; ^e^—patients with elevated TPO-ab and hypothyroidism; ^f^—children after the end of the randomized controlled trial; ^g^—TSH in euthyroid group; ^h^—all patients enrolled into the study; ^i^—previously reported control group; “+”—presence of clinically significant values of antibodies; “−“—lack of antibodies/not performed in the study.

**Table 2 ijms-26-00029-t002:** HT course in group with DS and non-DS group—comparison.

Non-DS Population	Population with DS
The onset of HT in later years	The onset of HT begins earlier
Euthyroidism as the most common first presentation	Subclinical hypothyroidism as the most common first presentation
Less susceptible to GD conversion	Are more susceptible to GD conversion
Correlated with female gender	Is not correlated with gender

DS—Down syndrome; HT—Hashimoto’s thyroiditis; GD—Graves’ disease.

**Table 3 ijms-26-00029-t003:** GD occurrence in patients with DS.

Author, Year	N Study Group of Patients with DS	N Control Group	Age (Range or Mean)	GD	% of GD	TSH UI/mL	fT4	Anti-TSHR/TRAb
Goday-Arno et al., 2009 [22]	1832	-	10.9–28.9 years	12	0.65%	Undetectable ^a^	63.7 pmol/L	+
Tüysüz et al., 2001 [96]	320	-	5 days–10 years	0	0%	-	-	-
Calcaterra et al., 2020 [98]	91	-	12.34 ± 8.37 years	4	4.4%	-	-	+
Nurcan Cebeci et al., 2024 [95]	161 ^b^	-	10.6 ± 4.5 years	13 ^c^	8%	<0.01 mU/L ^c^	50.2 ± 18.7 pmol/L	+

DS—Down syndrome; GD—Graves’ disease. ^a^—In patients with GD; ^b^—patients with GD with and without DS; ^c^—patients with DS with GD.

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
