# Peer review of "Autoimmune Thyroid Disease in Patients with Down Syndrome—Review"

_ijms, 2024, doi:10.3390/ijms26010029_

Round 1
Reviewer 1 Report
Comments and Suggestions for Authors
Authors provide an interesting and comprehensive review , easy to read, comprisins molecular data on Hashimoto Thyroiditis and Graves Disease and comparisons with non DS groups. The mansucript is easy to read and well organised
These are my suggestions:
As the paper' main objective is to focus on auto immune thyroid disease, I do not understant why all other auto immune disorders related to Down syndrome are also discussed.
They could also provide a table related to TSH an anti TPO Ab and Anti RSH R antibodies in Down syndrome patients compared to patients with autoimmune thyroiditis without Down syndrome to precise whether differences.
Authors could also cite this manuscript related to the subject.
https://doi.org/10.1038/s41586-023-05736-y
Reviewer 2 Report
Comments and Suggestions for Authors
The article concerns one of the most common genetic defects, which confirms the need for a thorough understanding of the pathophysiology of the disease and its accompanying diseases.
The article has been improved, and immunological pathophysiology has been thoroughly discussed.
The accompanying diseases, including immunodeficiencies, which are rarely thought of, have been explained.
There are individual typos and punctuation errors, e.g. in the table in the columns there are different sizes of letters in the initial words.
The words "level" should be changed to the words concentration.
In order to improve the manuscript, I propose to verify and improve the following points:

Reviewer 3 Report
Comments and Suggestions for Authors
Dear Authors,
The manuscript "Autoimmune Thyroid Disease in Patients with Down Syndrome," ijms-3344264, which reviews current knowledge of autoimmune thyroid disease in patients with Down Syndrome, is written in fluent English, and contains some interesting observations. However, none of the listed co-authors has auto citations in the field of Down Syndrome, and furthermore, two co-authors (Weronika Szybiak-Skora and Wojciech Cyna) have only one auto citation, which is again a review paper on the development of autoimmune thyroiditis, not an original article. Since this should be a review, the named authors lack sufficient competence and knowledge in the field, so they are not eligible to be the authors of this paper.
This publication is the only auto-citation of the authors:
Cyna, W.; Wojciechowska, A.; Szybiak-Skora, W.; Lacka, K. The Impact of Environmental Factors on the Development of Auoimmune Thyroiditis-Review. Biomedicines 2024, 12(8), 1788. doi: 10.3390/biomedicines12081788.
The second major remark is the structure of the paper. The discussion section should include a comparison of your results with published data. As this is a review, what exactly are the results of this study? Are they presented in Table 1 and Figure 4? If so, why are they placed into the discussion section?
Third, please explain all the factors in Table 1. Are the presented years, average/range ages of DS patients, the control group, or the total sample? What does the presented p-value refer to? For the valid comparison, each study presented should include the % of HT in the control group, as well as the average age of HT.
Minor:
- In the abstract, it is written: “According to previous results, collected in our study.” Please clarify this sentence. Are the results from your previously published data (if so, then explain which one) or just say, “according to published data…”.
- It is better if the keywords do not overlap the words from the title.
- Figure 4 is insufficient as the data are already presented in Table 1.
- Each table/figure should include the explanation of all abbreviations used in the table/figure.
- Please correct ref. no. 108: in all publications are solely those of the individual author(s) and contributor(s) and not of MDPI and/or the editor(s). MDPI and/or the editor(s) disclaim responsibility for any injury to people or property resulting from any ideas, methods, instructions or products referred to in the content.
Round 2
Reviewer 3 Report
Comments and Suggestions for Authors
Dear authors,
As long as your manuscript can contribute to the understanding of thyroid abnormalities in patients with Down syndrome, self-references are a requirement for writing the review.
I’m sorry, but none of you have references in the field of Down Syndrome. Furthermore, without previously published original papers in the scope of this review, Weronika Szybiak-Skora and Wojciech Cyna, are not eligible to be the authors of this paper. You should perform your planned study, publish the results, and then you may start writing the reviews.
In addition, among the provided references (doi:10.3390/ph16091320, doi:10.3390/biomedicines1108229, doi:10.5114/ada.2023.133457, doi:10.3390/ph17060728, doi:10.3390/biomedicines12081788), there is only one original paper, which relates to the patients with acne vulgaris. From the rest, there are two meta-analyses, one review, and one reference that doesn't even include the names of the authors (Mayer-Pickel, K.; Nanda, M.; Gajic, M.; Cervar-Zivkovic, M. Preeclampsia and the Antiphospholipid Syndrome. Biomedicines 2023, 11, 2298. https://doi.org/10.3390/biomedicines11082298)